# FedDID: Discrepancy-Informed Distillation to Target Personalization *and* Generalization in Federated Learning

## Abstract

Statistical heterogeneity poses a central challenge in federated learning (FL), degrading both local personalization through client class imbalance and global generalization through unstable knowledge retention across rounds. Prior work often treats these goals separately, yielding generic FL methods optimized for global performance and personalized FL methods tailored to local models; even recent approaches that consider both typically optimize them with distinct objectives. We observe that global-model regularization is a shared structure across both paradigms and can be leveraged to pursue both goals within a single mechanism. We propose **Federated Discrepancy-Informed Distillation (FedDID)**, which jointly promotes personalization and generalization by adaptively aligning local and global models via classwise knowledge distillation weighted by prediction-confidence discrepancies, without notable computational overhead. We provide theoretical motivation and show strong empirical performance under label heterogeneity, achieving the best overall balance across datasets by combining high global accuracy with low forgetting alongside strong local accuracy. On CIFAR-10 in particular, FedDID improves global accuracy by 17% over the next best-performing baseline while remaining competitive with the local performance of dedicated personalization methods.

## 1 Introduction

Federated learning (FL) trains a shared global model by aggregating client updates over multiple communication rounds, without exchanging raw data McMahan et al. (2017). FL is commonly assessed by *generalization* on a held-out global test set and *personalization* on each client's local test data McMahan et al. (2017); Zhang et al. (2023b); Firdaus et al. (2023). However, FL remains challenged by statistical heterogeneity, which can degrade performance at *both* the global and client levels.

Statistical heterogeneity occurs when each local client's data is drawn from heterogeneous data distributions, resulting in differing *label distributions*. This label heterogeneity causes local models to diverge from one another, ultimately creating a poor global model upon aggregation. Hence, poor global model performance is expressed not only in terms of its generalization accuracy, but also an inability to retain previously learned information (termed forgetting), shown in Figure 1.a through this decline in accuracy between communication rounds Lee et al. (2022); Shoham et al. (2019). In terms of personalization, class imbalance introduced by label heterogeneity hinders local models, increasing their risk of heavily overfitting to dominant classes as indicated by Figure 1.b.

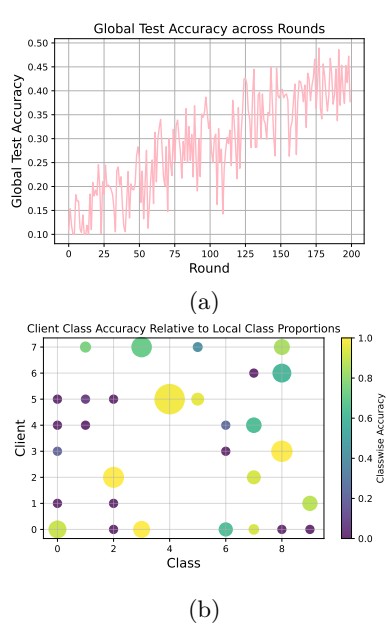

Figure 1: (**a**) Forgetting in FL and (**b**) Local overfitting to dominant classes

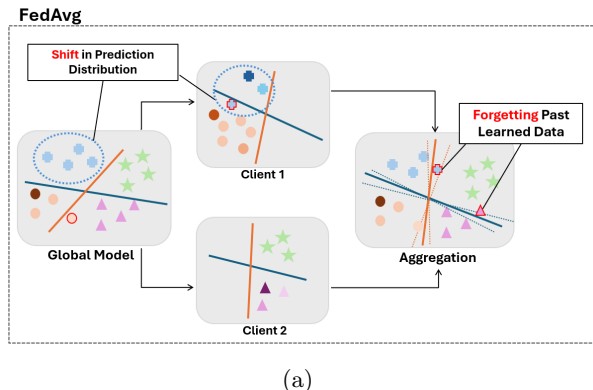 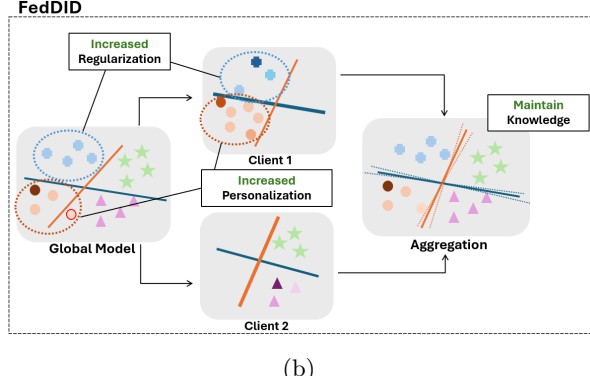

|  |  |
|---|---|
| (a) | (b) |

Figure 2: **(a)** FedAvg allows for local models to freely personalize to their data with no penalization, oftentimes resulting in overfitting to majority classes and forgetting global knowledge. **(b)** When faced with large shifts in prediction probabilities (indicated by different hues), FedDID enforces stronger regularization for these classes. Additionally, when prediction probabilities between the local and global model are more similar, regularization is decreased, allowing for increased personalization.

Even though label heterogeneity proves to be a challenge from both a generalization and personalization perspective, many prior works tend to focus on either the generalization *or* the personalization challenge separately. For instance, generic FL approaches focus on learning an overall global model that generalizes well despite label heterogeneity McMahan et al. (2017); Zhao et al. (2018). However, good generalization is accomplished at the cost of low performance on individual local clients Chen & Chao (2021). Personalized federated learning (pFL) instead creates local models that personalize well but sacrifice the global model's generalization capabilities, preventing new clients from seamlessly being integrated into the FL framework Firdaus et al. (2023); Tan et al. (2022a); Chen et al. (2024a).

While recent literature has begun to consider both personalization and generalization, these objectives are typically optimized separately, overlooking shared algorithmic foundations across generic and personalized FL approaches Chen & Chao (2021); Chen et al. (2024b). In particular, many methods across both lines of work incorporate some form of regularization with the global model to constrain local models from drifting too far from the global model. This regularization is commonly applied in generic approaches to reduce forgetting and in pFL approaches to reduce local overfitting Li et al. (2020; 2021b); Lee et al. (2022). In particular, *knowledge distillation* remains a common approach for this regularization, where the global model's knowledge is transferred to local clients.

Knowledge distillation strategies, however, are typically applied with a uniform penalty across all classes and clients, disregarding local relevance or any meaningful statistics Lee et al. (2022); Yashwanth et al. (2023). This uniform weighting, as a result, can present many challenges to the personalization and generalization goals. For instance, a client may possess local data that is poorly represented by the global model. Hence, too high a penalty on this regularization can directly hinder personalization. However, if the penalty is weighted too low, local models will diverge significantly from the global model, risking forgetting at the global level in addition to local overfitting. These limitations suggest that global regularization should instead adapt to each client's classwise understanding relative to the global model. For instance, overly confident local predictions can indicate overfitting to client data, while reduced confidence on globally well-learned classes can signal erosion of shared knowledge.

Motivated by these insights, we propose an adaptive, classwise penalty that turns shared regularization into a mechanism that supports both generalization and personalization. We find correlational evidence that drift in class prediction confidence between the global and local models on local training data can both flag classes at risk of being forgotten and reflect beneficial personalization. Based on this, we introduce FedDID (Figure 2), a knowledge distillation based FL method that uses this drift to adaptively weight local global alignment. We further modify aggregation to account for local label diversity and model quality,

strengthening global generalization. Across multiple image classification benchmarks, FedDID achieves the best balance overall, combining high global accuracy with low forgetting while maintaining competitive local performance. It also ranks among the top methods in worst-client accuracy, highlighting improved reliability across heterogeneous clients, and it maintains strong performance as the number of clients scales up. Finally, FedDID adds negligible communication overhead beyond FedAvg. Overall, FedDID is most effective under severe heterogeneity and remains robust as task difficulty increases.

Our main contributions can be summarized as follows:

1. We identify the drift in class prediction confidence between the local and global model as an informative statistic for identifying classes likely to be forgotten and for promoting personalization

2. We leverage this statistic to jointly tackle the generalization and personalization challenge, where each local client faces adaptive regularization

3. We propose a model aggregation strategy to further enhance global model generalization, where local model quality and label diversity inform aggregation weights

## 2 Related Works

### 2.1 Generalization and Personalization in FL

Many prior approaches separate the tasks of generalization and personalization. For instance, many prior works propose methods to enhance generalization when faced with label heterogeneity Li et al. (2019); Karimireddy et al. (2020); Acar et al. (2021); Yashwanth et al. (2023). For instance, FedProx Li et al. (2020) is a generic FL algorithm that adds a proximal term to the local training stage of each local client to promote consistency and regularization with the global model. Other studies attempt to target label heterogeneity from the perspective of mitigating forgetting, such as FedCurv, which compels local models towards finding a shared optimum Shoham et al. (2019). Other categories of FL approaches attempt to enhance personalization when faced with label heterogeneity. These approaches are denoted as personalized FL (pFL) methods, which tackle the heterogeneity challenge by developing tailored local models Fallah et al. (2020); T Dinh et al. (2020); Li et al. (2021b); Zhang et al. (2023b). pFL methods like FedALA Zhang et al. (2023b), for instance, selectively downloads information from the global model. Regularization with the global model also occurs in pFL approaches, seen through methods like Ditto Li et al. (2021b).

Recently, literature has focused on addressing both generalization and personalization simultaneously Chen & Chao (2021); Oh et al. (2021). A representative example is FedBABU, which explicitly decouples the two by first learning shared representations and then finetuning a personalized head Oh et al. (2021). However, this design still frames generalization and personalization as separate stages with competing goals. Instead, we show they need not be decoupled. By exploiting informative statistics available during local training, we directly optimize for global transfer and client adaptation within a single objective, enabling simultaneous gains without a two-stage training pipeline.

### 2.2 Knowledge Distillation in FL

A common form of regularization present within FL approaches is knowledge distillation, a mechanism to imbue a client's local model with the global model's knowledge by matching the logits outputted by both models Hinton (2015). Within these approaches, FedNTD Lee et al. (2022) uses knowledge distillation to preserve the global model's perspective on not-true classes to alleviate forgetting. Wu et al. (2022) also uses knowledge distillation via a mutual distillation loss. In terms of similarity to our work, FedHKD Chen et al. (2023) uses knowledge distillation with soft predictions to target both generalization and personalization, though this distillation is performed uniformly. Yashwanth et al. (2023), on the other hand, uses knowledge distillation in a semi-adaptive manner, with the weight being influenced by the global model's entropy. Our work instead uses knowledge distillation in a fully adaptive manner, informed in a classwise fashion by the discrepancy in prediction confidence between the local and global model.

## 3 Motivation

In this section, we aim to motivate relying on the drift in prediction confidence between the local and global model for informing knowledge distillation by relating this quantity to both (1) classwise forgetting at the global level and (2) clients' personalization capabilities. We accomplish this by conducting empirical studies on the CIFAR-10 dataset, where clients receive highly heterogeneous data partitions (partitioned via a Dirichlet distribution of $\alpha = 0.1$). Now, let $\mathcal{J}$ be the set of all possible classes. Then, for each local client $k$ sampled during a given communication round, we compute the absolute drift in prediction confidence as $|\Delta\mathcal{U}_j| = |\mathcal{U}_j(p_g) - \mathcal{U}_j(p_k)|$ for all $j \in \mathcal{J}$ classes for $(x, y)$ in the client's training dataset $\mathcal{D}_k$, where

$$\mathcal{U}_j(p) = -\mathbb{E}_{x \sim \mathcal{D}_k}[p(y = j) \log(p(y = j))] \tag{1}$$

for some set of given prediction probabilities $p$. Hence, if probabilities corresponding to a class are more confident, i.e. $p(y = j) \to 1$ or $p(y = j) \to 0$, $\mathcal{U}_j(p)$ will be lower. On the other hand, classes with more uncertainty result in a higher $\mathcal{U}_j(p)$. In this scenario, $\mathcal{U}_j(p_g)$ and $\mathcal{U}_j(p_k)$ represent the degree of confidence the global model and local model have about class $j$ on the client's dataset, respectively. We present additional experimental details and plots in Appendix F.6. We note that these correlation studies are meant as an *empirical motivating example*, not evidence that confidence drift fully explains forgetting or personalization. Since these effects are multi-factorial in heterogeneous FL, we do not expect any single scalar to be determinative; rather, we show that confidence drift is a consistent, actionable signal for deciding when and where to emphasize distillation.

### 3.1 Correlation to Forgetting

To test whether confidence drift relates to global classwise forgetting, we correlate $|\Delta\mathcal{U}_j|$ with the change in global classwise accuracy for classes absent from a client's local distribution, since these classes are most vulnerable to forgetting. Averaging the Pearson correlation coefficient across clients for each round yields an average correlation of $r \approx -0.42$ across rounds. While this magnitude indicates only a moderate association (expected given the many contributors to global forgetting like sampling and optimization noise), it suggests that confidence drift captures a meaningful component of the forgetting process. Notably, this result indicates that large changes in confidence of local model predictions relative to global predictions can correlate to higher risk of forgetting. Even if the local model becomes more confident than the global model in its predictions, this can be harmful to global knowledge retention, potentially due to overconfidence of the local model or representation collapse. These results suggest that monitoring shifts in prediction confidence provides a useful early-warning signal for identifying classes that may be at higher risk of being forgotten, allowing for targeted intervention on these classes during training.

### 3.2 Correlation to Personalization

We next examine whether confidence drift correlates with personalization by computing, for each sampled client $k$, the correlation between $|\Delta\mathcal{U}_j|$ and the current client $k$'s local classwise accuracy for every class that is present in the local data distribution. For stability, we use a milder heterogeneity setting (Dirichlet $\alpha = 0.5$).

We compute the average Pearson correlation coefficient across clients for each round, finding that the average Pearson correlation coefficient across rounds is $r \approx -0.51$. This result also suggests a moderate relationship between absolute shifts in prediction confidence and resulting personalization performance, where the local model becoming more confident relative to the global model, even on data that's within its own dataset, can indicate *worse* personalization performance.

Even with a less harsh label heterogeneity setting, class imbalance is still prevalent. Hence, we also compute the correlation between $|\Delta\mathcal{U}_j|$ and the current client $k$'s local classwise accuracy for every class that (1) appears in the local distribution and (2) has fewer than 40% of the instances of the most frequent class. This average Pearson correlation coefficients across clients for each round then becomes $r \approx -0.55$, indicating that performance on less prevalent classes is tied to the discrepancy in prediction confidence. Overall, confidence drift appears especially informative for monitoring personalization on minority local classes.

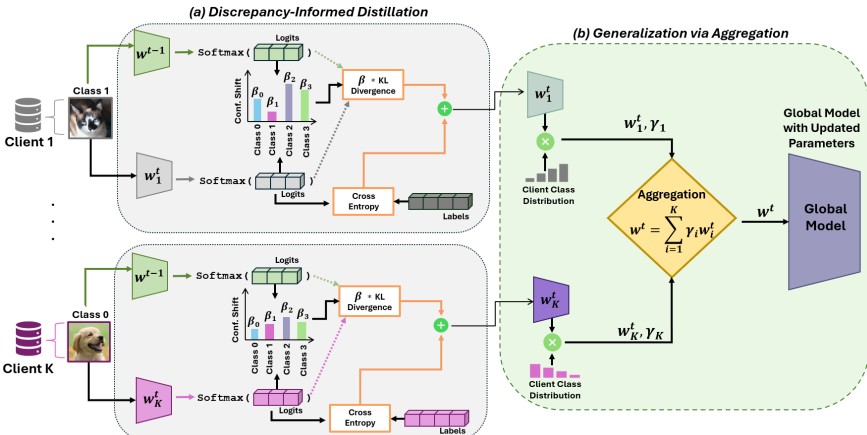

Figure 3: FedDID introduces **(a)** Discrepancy-Informed Distillation via adaptive regularization with the global model based on shifts in local prediction confidence and **(b)** enhanced generalization by weighting local clients based on their label quantity and model quality.

## 4 Methodology

### 4.1 Problem Formulation

In standard FL setups McMahan et al. (2017), we assume access to $K$ local clients. Each client $k$ has its own local dataset, which we denote as $\mathcal{D}_k$ of size $n_k$. At each communication round $t$, we randomly sample a portion of the total clients $K^t$. The current global model parameters $w^{t-1}$ are then distributed to each of these sampled clients. Each sampled local client $k \in K^t$ of dataset size $n_k$ updates its model parameters $w$ by using the following objective:

$$F_k(w) = \frac{1}{n_k} \sum_{i=1}^{n_k} \ell(x_i, y_i; w) \tag{2}$$

where $\ell$ represents the loss (typically cross-entropy $\ell = \ell_{\text{CE}}$) on the $k$-th client's local example $(x_i, y_i)$ from its local dataset $\mathcal{D}_k$. The global objective across sampled clients optimized during training is then defined as $F(w) = \sum_{k \in K^t} \gamma_k F_k(w)$. where $\gamma_k$ represents the weight each local client is assigned in the global averaging step. In the case of FedAvg, $\gamma_k = \frac{n_k}{\sum_{j \in K^t} n_j}$, i.e. the size of the respective local dataset impacts the weight assigned to the local parameter McMahan et al. (2017). After local optimization, the server updates the global parameters at round $t$ via weighted averaging $w^t = \sum_{k \in K^t} \gamma_k w_k^t$.

### 4.2 Formulation of FedDID

Our proposed method FedDID (Figure 3) is made up of two distinct components: (1) discrepancy-informed distillation within the local training stage and (2) a weighted aggregation strategy to take into account local label diversity and model quality. We describe each of the components below and provide the overall algorithm in Appendix A.

#### 4.2.1 Discrepancy-Informed Distillation

In FL, knowledge distillation matches the logits outputted by both a local and global model on a client's dataset. However, knowledge distillation is traditionally applied uniformly across all clients and classes. Statistical heterogeneity commonly results in local models specializing heavily to classes present in abundant amounts, while failing to perform well on uncommon classes. This over-specialization not only harms the overall personalization, but it results in global information about specific classes being forgotten as well. Hence, to mitigate these challenges, we apply an adaptive weight to each class that participates in the knowledge distillation process, relying on local and global statistics (Figure 3.a). The novelty in FedDID

is that the distillation pressure is not fixed; it is instead allocated where forgetting and overfitting is most likely. By using the confidence drift $|\Delta\mathcal{U}_j|$ as a per-class signal, we bias alignment toward at-risk classes (i.e., classes for which the local and global models exhibit the largest confidence mismatch), rather than regularizing all classes equally. This yields a fine-grained retention–adaptation control that adapts to each client's non-IID label support without requiring manual tuning of per-dataset or per-client regularization strengths.

With our method, the $k$-th client minimizes the following objective:

$$\min_w h_k(w) = F_k(w) + R_k(w) \tag{3}$$

$$\text{with } F_k(w) = \mathbb{E}_{x\sim\mathcal{D}_k}[\ell_{\text{CE}}(\hat{y}, y)] \tag{4}$$

$$R_k(w) = \mathbb{E}_{x\sim\mathcal{D}_k}[\sum_{j\in\mathcal{J}} \beta_{k,j}\, \ell_{\text{KL}}(p_{k,j}, p_{g,j})] \tag{5}$$

Hence, the local objective consists of a cross-entropy loss $\ell_{\text{CE}}$ between the local client's predictions on its dataset $\hat{y}$ and the actual target values $y$. The second term in the local objective is the knowledge distillation process, consisting of a KL divergence term to match the softmax probability predictions of the local model $p_k$ with those of the global model $p_g$ on the local dataset for each of the $j \in \mathcal{J}$ classes. Additionally, this alignment for client $k$ is adaptively weighted by a confidence regularization factor $\beta_{k,j}$, described below as:

$$\beta_{k,j} = \begin{cases} 0 & \text{if } y = j, \\ 1 + |\Delta\mathcal{U}_j| & \text{otherwise} \end{cases}$$

where $|\Delta\mathcal{U}_j|$ refers to the drift in prediction confidence between the current local and global model regarding class $j$ on the *local training data*. In other words, if the local model's predictions drift too heavily from the global model, this drift is penalized via this adaptive weight. This weighting prevents the local model from both becoming too confident on a particular class with respect to the global model's predictions, as well as becoming too uncertain. Additionally, the true class is assigned a weight of 0 to not penalize divergence on the correct class, similar to FedNTD Lee et al. (2022). We present an analysis of our choice of $\beta$ in Section 4.3. Importantly, with this statistic, FedDID is as low-cost as other knowledge distillation methods, since it only requires a difference measurement between the global and local predicted probabilities.

### 4.2.2 Model Aggregation

To further promote generalization, we provide a new model aggregation strategy that takes into account (1) client label diversity and (2) client model quality, as shown in Figure 3.b. Specifically, we accomplish this by weighting each sampled client $k \in K^t$ with the corresponding label distribution $\mathcal{Y}_k$ in proportion to its training accuracy $\mathcal{A}$ and its label diversity. Hence, for round $t$ we assign $\bar{\gamma}_k$ as the calculation for client $k$'s weight as follows:

$$\bar{\gamma}_k = \bar{J}\mathcal{A}_k^t \tag{6}$$

where $\bar{J} = \frac{1}{J}\sum_{j\in\mathcal{J}} \mathbf{1}(j \in \mathcal{Y}_k)$ denotes the proportion of the full class set $\mathcal{J}$ that is present on client $k$, with $J = |\mathcal{J}|$ representing the total number of classes. We can write the total model aggregation weights for sampled clients $k \in K^t$ as $\gamma_k = \frac{\bar{\gamma}_k}{\sum_{j\in K^t} \bar{\gamma}_j}$, where $\sum_{k\in K^t} \gamma_k = 1$. This weighting prioritizes clients whose updates are both *well-fit* and *informative across more labels* (high $\bar{J}$), thereby reducing dominance from highly skewed clients and encouraging a global update that better reflects broader class knowledge. By doing this, the global model is regularized by higher-performing local models, enhancing the overall generalization capabilities of the global model. This reliability-diversity weighting is intentionally designed for the federated setting: it provides a principled mechanism to discourage over-confident but highly skewed client updates and to promote updates that are both accurate and label-covering. Furthermore, our server only receives a single scalar score per client ($\bar{\gamma}_k$), which is computed entirely on-device. This scalar acts only as a coarse aggregation weight and does not expose client per-class counts or any sample-level information, maintaining client privacy.

### 4.3 Theoretical Analysis

In this section, we discuss how confidence discrepancies between the local and global model can inform the knowledge distillation process to jointly target generalization and personalization from a theoretical lens. Additionally, we analyze FedDID's convergence in Appendix B.

#### 4.3.1 Generalization via Reduced Forgetting

We now wish to provide a theoretical grounding as to why our adaptive weight jointly targets the generalization and personalization goals. Our formulation of the loss (Equation 3) and corresponding weighting aims to improve generalization by increasing local alignment with the global model on classes that pose a risk to being forgotten. Hence, to understand how our adaptive weight achieves this goal, we begin by introducing a definition of forgetting to quantify this information loss.

**Definition 1 (Forgetting)** *Given a local dataset $(X, Y) \sim \mathcal{D}_k$, let $I_g(X; Y)$ represent the global model's mutual information and $I_k(X; Y)$ denote the local model's mutual information between local inputs and labels. We can then determine that some amount of forgetting $\mathcal{F}$ has occurred if $\mathcal{F} = I_g(X; Y) - I_k(X; Y) > 0$.*

With this definition in place, we can now show that the amount of forgetting caused by training on client $k$ can be bounded by a function of the change in prediction confidence between the local and global model.

**Lemma 1 (Bounding Forgetting)** *Using Definition 1, the total amount of forgetting $\mathcal{F}$ can be bounded as $\mathcal{F} \leq \Delta\mathcal{H}(Y) + \log(J)$, where $\Delta\mathcal{H}$ represents the change in entropy between the global and local model and $J$ represents the total number of classes. Defining $h_j(p) = -p(y = j)\log(p(y = j))$, we can rewrite $\Delta\mathcal{H}(Y) = \sum_{j \in \mathcal{J}} \Delta h_j$. It then follows naturally that*

$$\Delta\mathcal{U}_j = \mathbb{E}_{x \sim \mathcal{D}_k}[\Delta h_j] < \mathcal{F} \tag{7}$$

*With this relationship, we can relate our definition of forgetting to $\Delta\mathcal{U}_j$ via the bound*

$$\mathbb{E}_{x \sim \mathcal{D}_k}[\mathcal{F}] \leq \sum_{j \in \mathcal{J}} \Delta\mathcal{U}_j + \log(J) \leq \sum_{j \in \mathcal{J}} |\Delta\mathcal{U}_j| + \log(J) \tag{8}$$

In summary, Lemma 1 states that the average amount of forgetting that occurs due to training on client $k$ is upper-bounded by the cumulative prediction confidence shift across all classes. Hence, this result suggests that large shifts in prediction confidence can contribute to forgetting at the global level and further motivates the use of including $|\Delta\mathcal{U}_j|$ to inform regularization on specific classes. A full proof is provided in Appendix C.1.

#### 4.3.2 Potential for Personalization Enhancement

We next provide an interpretive analysis showing when the KD term can contribute additional descent in the personalization objective, intended to characterize the mechanism by which FedDID may support personalization. To accomplish this, we introduce two assumptions and derive the influence of our combined loss on the pure local objective, as measured by cross-entropy.

**Assumption 1 (Bounded Regularization)** *For $w \in \mathbb{R}^d$, let $\{\beta_{k,j}\}_{j=1}^J$ be the classwise weights and define $\beta_k = \max_j \beta_{k,j}$. Then, the gradient of the regularization term for an arbitrary client can be upper-bounded by $||\nabla R_k(w)|| \leq \beta_k ||\nabla F_k(w)||$.*

**Assumption 2 (Bounded Alignment)** *For $w \in \mathbb{R}^d$, the dissimilarity between the regularization and local objective gradient can be lower-bounded by $\langle \nabla R_k(w), \nabla F_k(w) \rangle \geq -\alpha ||\nabla F_k(w)||^2$, where $\alpha \in [0, 1)$.*

**Lemma 2 (Personalization Gain)** *When the local loss solely consists of a cross-entropy term, the effect of a local update on personalization between rounds $t$ and $t+1$ can be approximated by $F_k(w_k^{t+1}) - F_k(w_k^t) \approx -\eta ||\nabla F_k(w_k^t)||^2$. When adding in our KL term, the descent progress of this personalization objective for*

*client k is then denoted as $F_k(w_k^{t+1}) - F_k(w_k^t) \approx -\eta(||\nabla F_k(w_k^t)||^2 + \langle \nabla F_k(w_k^t), \nabla R_k(w_k^t) \rangle)$. Thus, knowledge distillation provides additional descent $d_k$ in the pure local direction via*

$$d_k = \eta \langle \nabla F_k(w_k^t), \nabla R_k(w_k^t) \rangle \tag{9}$$

*Utilizing Assumptions 1 and 2, we note that $d_k$ can be bounded as*

$$-\eta\alpha||\nabla F_k(w)||^2 \le d_k \le \eta\beta_k||\nabla F_k(w)||^2 \tag{10}$$

*for $\alpha \in [0, 1)$ and $\beta_k = \max_j \beta_{k,j}$. Even when some degree of gradient misalignment is present, yielding the worst-case bound for $d_k$, Equation 3 still yields some personalization gain of at least $-\eta(1-\alpha)\|\nabla F_k(w_k^t)\|^2$. If $\langle \nabla F_k, \nabla R_k \rangle > 0$, the KL term yields enhanced personalization relative to cross-entropy alone.*

Enhanced personalization is typically true if (1) the KL term does not dominate the overall loss and (2) the KL and cross entropy term gradients are not opposed in such a way that hinders personalization. In addition to Assumption 1, this first condition is satisfied by our definition of $\beta_{k,j}$ stating that $\beta_{k,j} = 0$ if $j$ corresponds to the true class, or it is weighted proportionally to $1 + |\Delta\mathcal{U}_j|$, which is naturally bounded between $[1, 2]$, guaranteeing that the KL term does not dominate over the overall optimization. The second condition is captured directly by Lemma 2 via the inner product term. Even if KL and cross entropy gradients have some degree of misalignment, the bounded alignment assumption ensures that the overall personalization descent remains positive in the worst case. Therefore, our formulation of the loss ensures consistent personalization improvement across rounds. A full proof is provided in Appendix C.2.

## 5 Evaluation

### 5.1 Datasets and Baselines

We evaluate FedDID on three small-class datasets ($\le 10$ classes): CIFAR-10, BloodMNIST, and OrganCM-NIST Krizhevsky et al. (2009); Yang et al. (2023). We also evaluate on two large-class datasets: CIFAR-100 and TinyImageNet Krizhevsky et al. (2009); Le & Yang (2015). We use 100 clients for CIFAR-10/100, 200 for TinyImageNet, and fewer clients for the smaller medical datasets (20 for BloodMNIST, 30 for OrganCM-NIST). We compare against generic FL baselines (FedAvg McMahan et al. (2017), FedAvg+ASD Yashwanth et al. (2023), FedProx Li et al. (2020), MOON Li et al. (2021a), FedCurv Shoham et al. (2019), FedNTD Lee et al. (2022)), personalization methods (FedAS Yang et al. (2024), FedALA Zhang et al. (2023b), Ditto Li et al. (2021b), FedPer Arivazhagan et al. (2019), FedKD Wu et al. (2022)), and joint generalization-personalization approaches (FedBABU Oh et al. (2021), FedHKD Chen et al. (2023)).

### 5.2 Experimental Design and Details

We partition via a Dirichlet distribution $Dir(\alpha)$ as done in Li et al. (2021a). This partitioning involves allotting a portion of the samples in class $j$ to be assigned to client $k$. We set $\alpha = 0.1$ for all datasets except TinyImageNet (where we set $\alpha = 0.2$) in order to generate high data heterogeneity. We set the number of communication rounds equal to 200. At each round, we sample 10% of the total clients (5% for TinyImageNet). Additional training and evaluation details can be found in Appendix E.

For evaluation, we measure generalization using global test accuracy averaged across communication rounds and three seeds. We also wish to understand the global model's capacity for retaining past learned information. Hence, we adapt the concept of Negative Flip-Rate (NFR) Yan et al. (2021) to measure forgetting on a test set of size $N$:

$$\text{NFR} = \frac{1}{N} \sum_{i=1}^{N} \mathbf{1}(\hat{y}_i^{t-1} = y_i, \hat{y}^t \ne y_i) \tag{11}$$

where for sample $i$, $\mathbf{1}(\hat{y}_i^{t-1} = y_i, \hat{y}^t \ne y_i)$ equals one if the prediction of this sample $\hat{y}_i^t$ is now incorrect at the current round $t$ after originally being correct in the prior round $t-1$.

To assess personalization, we report the average local test set accuracy attained by local models across communication rounds. We also calculate the average local classwise accuracy to take into account performance

Table 1: Global and local results for datasets with smaller number of classes over 3 seeds.

| Dataset | Strategy | Global | | Local | | Balance ↑ |
|---|---|---|---|---|---|---|
| | | Acc ↑ | NFR ↓ | Acc ↑ | Class Acc ↑ | |
| CIFAR-10 | FedAvg McMahan et al. (2017) | 0.287 ±0.101 | 0.154 ±0.048 | 0.625 ±0.079 | 0.552 ±0.080 | 0.456 |
| | FedAvg+ASD Yashwanth et al. (2023) | 0.387 ±0.090 | 0.024 ±0.014 | 0.569 ±0.064 | 0.539 ±0.068 | 0.478 |
| | FedCurv Shoham et al. (2019) | 0.338 ±0.101 | 0.156 ±0.041 | 0.651 ±0.076 | 0.583 ±0.078 | 0.495 |
| | MOON Li et al. (2021a) | 0.286 ±0.099 | 0.152 ±0.047 | 0.623 ±0.079 | 0.550 ±0.079 | 0.455 |
| | FedProx Li et al. (2020) | 0.382 ±0.084 | 0.131 ±0.042 | 0.651 ±0.074 | 0.585 ±0.077 | 0.516 |
| | FedNTD Lee et al. (2022) | 0.433 ±0.113 | 0.092 ±0.042 | 0.682 ±0.077 | 0.622 ±0.080 | 0.558 |
| | FedAS Yang et al. (2024) | 0.206 ±0.038 | 0.152 ±0.037 | **0.791** ±0.040 | **0.791** ±0.040 | 0.499 |
| | FedALA Zhang et al. (2023b) | 0.375 ±0.056 | 0.128 ±0.029 | 0.661 ±0.041 | 0.592 ±0.043 | 0.518 |
| | Ditto Li et al. (2021b) | 0.364 ±0.079 | 0.127 ±0.028 | 0.784 ±0.039 | 0.784 ±0.039 | 0.574 |
| | FedPer Arivazhagan et al. (2019) | 0.293 ±0.057 | 0.157 ±0.034 | 0.782 ±0.038 | 0.782 ±0.038 | 0.537 |
| | FedKD Wu et al. (2022) | 0.196 ±0.022 | 0.138 ±0.032 | 0.594 ±0.043 | 0.517 ±0.044 | 0.395 |
| | FedBABU Oh et al. (2021) | 0.389 ±0.080 | 0.132 ±0.042 | 0.668 ±0.070 | 0.605 ±0.075 | 0.529 |
| | FedHKD Chen et al. (2023) | 0.101 ±0.007 | **0.006** ±0.022 | 0.139 ±0.105 | 0.145 ±0.085 | 0.120 |
| | **FedDID (Ours)** | **0.505** ±0.110 | 0.082 ±0.028 | 0.706 ±0.074 | 0.646 ±0.080 | **0.605** |
| BloodMNIST | FedAvg McMahan et al. (2017) | 0.409 ±0.147 | 0.159 ±0.087 | 0.775 ±0.148 | 0.664 ±0.152 | 0.592 |
| | FedAvg+ASD Yashwanth et al. (2023) | 0.615 ±0.128 | **0.022** ±0.026 | 0.705 ±0.186 | 0.708 ±0.165 | 0.66 |
| | FedCurv Shoham et al. (2019) | 0.472 ±0.169 | 0.155 ±0.092 | 0.806 ±0.141 | 0.708 ±0.150 | 0.639 |
| | MOON Li et al. (2021a) | 0.426 ±0.153 | 0.157 ±0.087 | 0.787 ±0.143 | 0.677 ±0.150 | 0.606 |
| | FedProx Li et al. (2020) | 0.431 ±0.155 | 0.169 ±0.112 | 0.828 ±0.147 | 0.750 ±0.166 | 0.629 |
| | FedNTD Lee et al. (2022) | 0.589 ±0.177 | 0.101 ±0.084 | 0.849 ±0.120 | 0.770 ±0.142 | 0.719 |
| | FedAS Yang et al. (2024) | 0.315 ±0.074 | 0.193 ±0.068 | 0.773 ±0.128 | 0.685 ±0.135 | 0.544 |
| | FedALA Zhang et al. (2023b) | 0.516 ±0.113 | 0.175 ±0.065 | **0.899** ±0.068 | **0.845** ±0.081 | 0.708 |
| | Ditto Li et al. (2021b) | 0.450 ±0.127 | 0.159 ±0.056 | 0.794 ±0.120 | 0.706 ±0.126 | 0.622 |
| | FedPer Arivazhagan et al. (2019) | 0.325 ±0.084 | 0.198 ±0.067 | 0.782 ±0.125 | 0.699 ±0.135 | 0.554 |
| | FedKD Wu et al. (2022) | 0.426 ±0.102 | 0.160 ±0.059 | 0.811 ±0.103 | 0.709 ±0.109 | 0.618 |
| | FedBABU Oh et al. (2021) | 0.523 ±0.156 | 0.156 ±0.098 | 0.870 ±0.122 | 0.819 ±0.137 | 0.696 |
| | FedHKD Chen et al. (2023) | 0.159 ±0.026 | 0.092 ±0.048 | 0.482 ±0.127 | 0.368 ±0.110 | 0.321 |
| | **FedDID (Ours)** | **0.653** ±0.173 | 0.089 ±0.076 | 0.880 ±0.112 | 0.818 ±0.134 | **0.767** |
| OrganCMNIST | FedAvg McMahan et al. (2017) | 0.568 ±0.207 | 0.111 ±0.091 | 0.844 ±0.131 | 0.808 ±0.145 | 0.706 |
| | FedAvg+ASD Yashwanth et al. (2023) | 0.595 ±0.134 | **0.022** ±0.022 | 0.817 ±0.121 | 0.803 ±0.130 | 0.706 |
| | FedCurv Shoham et al. (2019) | 0.551 ±0.201 | 0.120 ±0.099 | 0.830 ±0.126 | 0.791 ±0.138 | 0.69 |
| | MOON Li et al. (2021a) | 0.559 ±0.203 | 0.112 ±0.092 | 0.840 ±0.131 | 0.805 ±0.145 | 0.7 |
| | FedProx Li et al. (2020) | 0.544 ±0.146 | 0.095 ±0.071 | 0.836 ±0.107 | 0.809 ±0.116 | 0.69 |
| | FedNTD Lee et al. (2022) | 0.618 ±0.175 | 0.073 ±0.073 | 0.868 ±0.109 | 0.841 ±0.121 | 0.743 |
| | FedAS Yang et al. (2024) | 0.371 ±0.085 | 0.188 ±0.069 | 0.810 ±0.087 | 0.776 ±0.100 | 0.591 |
| | FedALA Zhang et al. (2023b) | 0.523 ±0.107 | 0.116 ±0.050 | 0.884 ±0.065 | 0.867 ±0.078 | 0.703 |
| | Ditto Li et al. (2021b) | 0.564 ±0.131 | 0.113 ±0.046 | 0.841 ±0.094 | 0.813 ±0.106 | 0.702 |
| | FedPer Arivazhagan et al. (2019) | 0.415 ±0.096 | 0.170 ±0.058 | 0.818 ±0.082 | 0.784 ±0.095 | 0.616 |
| | FedKD Wu et al. (2022) | 0.328 ±0.083 | 0.137 ±0.047 | 0.731 ±0.091 | 0.660 ±0.100 | 0.53 |
| | FedBABU Oh et al. (2021) | 0.557 ±0.146 | 0.110 ±0.078 | 0.855 ±0.093 | 0.825 ±0.103 | 0.706 |
| | FedHKD Chen et al. (2023) | 0.161 ±0.045 | 0.062 ±0.047 | 0.348 ±0.129 | 0.298 ±0.124 | 0.255 |
| | **FedDID (Ours)** | **0.697** ±0.147 | 0.053 ±0.037 | **0.894** ±0.089 | **0.870** ±0.102 | **0.795** |

Table 2: Global and local results for datasets with a large number of classes over 3 seeds.

| Dataset | Strategy | Global | | Local | | Balance ↑ |
|---|---|---|---|---|---|---|
| | | Acc ↑ | NFR ↓ | Acc ↑ | Class Acc ↑ | |
| TinyImageNet | FedAvg | 0.182 ±0.049 | 0.052 ±0.009 | 0.144 ±0.039 | 0.138 ±0.037 | 0.163 |
| | FedAvg+ASD | 0.149 ±0.052 | 0.018 ±0.013 | 0.153 ±0.049 | 0.149 ±0.048 | 0.151 |
| | FedCurv | 0.161 ±0.041 | 0.050 ±0.008 | 0.130 ±0.034 | 0.124 ±0.032 | 0.145 |
| | MOON | 0.181 ±0.049 | 0.052 ±0.009 | 0.144 ±0.039 | 0.139 ±0.038 | 0.162 |
| | FedProx | 0.148 ±0.042 | 0.032 ±0.007 | 0.124 ±0.036 | 0.118 ±0.035 | 0.136 |
| | FedNTD | **0.201** ±0.052 | 0.021 ±0.004 | **0.187** ±0.051 | **0.183** ±0.052 | **0.194** |
| | FedAS | 0.005 ±0.000 | 0.000 ±0.001 | 0.006 ±0.003 | 0.006 ±0.002 | 0.005 |
| | FedALA | 0.090 ±0.021 | 0.027 ±0.005 | 0.084 ±0.022 | 0.078 ±0.020 | 0.087 |
| | Ditto | 0.090 ±0.021 | 0.027 ±0.005 | 0.085 ±0.023 | 0.079 ±0.021 | 0.087 |
| | FedPer | 0.070 ±0.015 | 0.026 ±0.005 | 0.070 ±0.017 | 0.065 ±0.016 | 0.07 |
| | FedKD | 0.056 ±0.024 | **0.016** ±0.007 | 0.048 ±0.016 | 0.042 ±0.014 | 0.052 |
| | FedBABU | 0.092 ±0.018 | 0.027 ±0.005 | 0.084 ±0.017 | 0.078 ±0.015 | 0.088 |
| | FedHKD | 0.005 ±0.001 | 0.000 ±0.000 | 0.007 ±0.006 | 0.007 ±0.005 | 0.006 |
| | **FedDID (Ours)** | 0.189 ±0.047 | 0.026 ±0.004 | 0.162 ±0.040 | 0.157 ±0.040 | 0.175 |
| CIFAR-100 | FedAvg | 0.251 ±0.078 | 0.064 ±0.013 | 0.286 ±0.067 | 0.285 ±0.068 | 0.268 |
| | FedAvg+ASD | 0.217 ±0.058 | 0.016 ±0.007 | 0.268 ±0.057 | 0.268 ±0.058 | 0.243 |
| | FedCurv | 0.240 ±0.073 | 0.062 ±0.012 | 0.270 ±0.064 | 0.270 ±0.064 | 0.255 |
| | MOON | 0.249 ±0.078 | 0.063 ±0.013 | 0.287 ±0.067 | 0.285 ±0.068 | 0.268 |
| | FedProx | 0.213 ±0.070 | 0.043 ±0.011 | 0.239 ±0.066 | 0.238 ±0.066 | 0.226 |
| | FedNTD | 0.272 ±0.081 | 0.027 ±0.005 | **0.312** ±0.075 | **0.311** ±0.076 | **0.292** |
| | FedAS | 0.039 ±0.012 | 0.032 ±0.010 | 0.113 ±0.028 | 0.113 ±0.028 | 0.076 |
| | FedALA | 0.177 ±0.033 | 0.036 ±0.006 | 0.199 ±0.024 | 0.199 ±0.024 | 0.188 |
| | Ditto | 0.144 ±0.042 | 0.040 ±0.007 | 0.175 ±0.037 | 0.175 ±0.037 | 0.159 |
| | FedPer | 0.100 ±0.023 | 0.044 ±0.008 | 0.145 ±0.021 | 0.144 ±0.021 | 0.122 |
| | FedKD | 0.131 ±0.030 | 0.033 ±0.006 | 0.136 ±0.016 | 0.136 ±0.016 | 0.133 |
| | FedBABU | 0.182 ±0.038 | 0.041 ±0.008 | 0.200 ±0.034 | 0.199 ±0.034 | 0.191 |
| | FedHKD | 0.010 ±0.001 | **0.001** ±0.003 | 0.026 ±0.014 | 0.026 ±0.013 | 0.018 |
| | **FedDID (Ours)** | **0.278** ±0.080 | 0.031 ±0.006 | 0.307 ±0.068 | 0.307 ±0.069 | **0.292** |

on underrepresented local classes. Finally, we wish to understand how each FL algorithm balances the generalization and personalization tasks. To this end, we devise a metric to measure this global-local balance as:

$$\text{Balance} = \frac{\bar{\mathcal{A}}_g + \bar{\mathcal{A}}_k}{2} \tag{12}$$

where $\bar{\mathcal{A}}_g$ and $\bar{\mathcal{A}}_k$ represent the average global and local test set accuracies respectively. This metric is our most important assessment, as it is geared towards understanding how each method addresses the personalization *and* generalization challenge simultaneously.

## 5.3 Experimental Results

### 5.3.1 Global and Local Performance Results

In Table 1, we compare FedDID against the described baselines on datasets with a smaller number of classes (10 or less). In Table 2, we compare FedDID on datasets with a larger number of classes (100 or greater). Overall, FedDID has the ability to maintain strong generalization and personalization capabilities, as indicated by the balance scores across each dataset. On datasets with smaller class sizes, FedDID surpasses all generic FL approaches' generalization accuracies, producing nearly a 17% increase in global test accuracy compared to the second-best algorithm on CIFAR-10.

We can also see that FedDID excels at maintaining past learned information, as indicated by the NFR. Methods like FedAvg+ASD report better NFR scores; however, their accuracy gains are overall limited compared to FedDID, implying these methods place too much emphasis on the regularization term. Instead, FedDID obtains the lowest NFR among all other baselines that *also* achieve strong generalization accuracy. For personalization on datasets with smaller class sizes, FedDID remains competitive with pFL approaches, producing the highest personalization performance of all non-pFL algorithms on the CIFAR-10 dataset. We also see that FedDID maintains strong classwise personalization performance, often beating or producing comparable results to pFL methods. Overall, Table 1 shows that FedDID produces the strongest generalization performance without sacrificing personalization, achieving the best

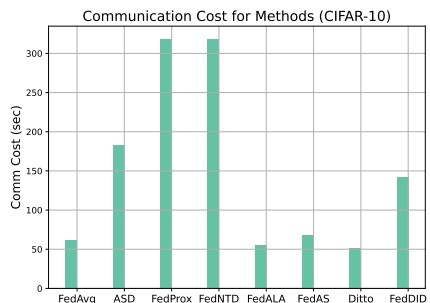

Figure 4: Communication cost

balance of both objectives. On datasets with a larger number of classes (Table 2), FedDID consistently demonstrates strong performance across global and local metrics. On CIFAR-100, FedDID ties as the top-performing method in terms of global-local balance. On TinyImageNet, FedDID ranks closely behind FedNTD, showing that its advantage narrows but remains competitive in larger class spaces. This setting is challenging due to unstable classwise knowledge alignment under extreme label sparsity, making large-class heterogeneous FL an important direction for future work. We also note that some TinyImageNet baselines show degenerate behavior, with near-chance global accuracy and zero NFR; these NFR values should not be interpreted as meaningful retention because the models failed to learn nontrivial task knowledge.

### 5.3.2 Communication Cost

We compare the average communication cost (wall-clock time) over 200 rounds of FedDID with FedAvg and other methods in Figure 4 on the CIFAR-10 dataset. As indicated by the figure, FedDID presents minor additional cost compared to FedAvg and is overall more efficient than other methods performing regularization with the global model during local training. Although some pFL methods incur lower wall-clock cost, FedDID provides stronger global generalization while still maintaining modest overhead. This suggests that FedDID's adaptive distillation mechanism improves performance without compromising scalability.

### 5.3.3 Performance Gaps Across Clients

Given that our proposed aggregation relies in part on local training accuracy (which can potentially overweight highly skewed clients), we compare average and worst-client local accuracy. As shown in Figure 5, FedDID maintains strong average accuracy while achieving the second-best worst-client accuracy, closely trailing FedALA (and FedNTD for CIFAR-100). This is especially notable because FedDID is *not* a dedicated pFL baseline like FedALA, showing that our method improves the global-local balance in a way that benefits not just the average client but also the most challenging clients.

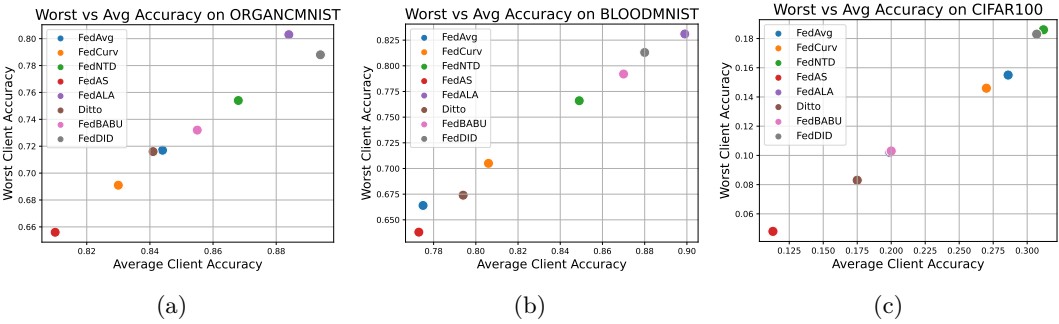

Figure 5: Average vs worst-client accuracy on **(a)** OrganCMNIST, **(b)** BloodMNIST, and **(c)** CIFAR-100

## 5.4 Ablation Studies

### 5.4.1 Varying Degrees of Label Heterogeneity

Figure 6 compares FedDID to five FL baselines under varying label heterogeneity (full results in Appendix F.7). FedDID remains robust across imbalance levels: under severe heterogeneity ($\alpha = 0.1$, Fig. 6.a) it achieves the best global generalization while its local models are second only to FedAS, and under milder heterogeneity ($\alpha = 0.5$, Fig. 6.c) it still yields the best global model while also outperforming pFL methods locally.

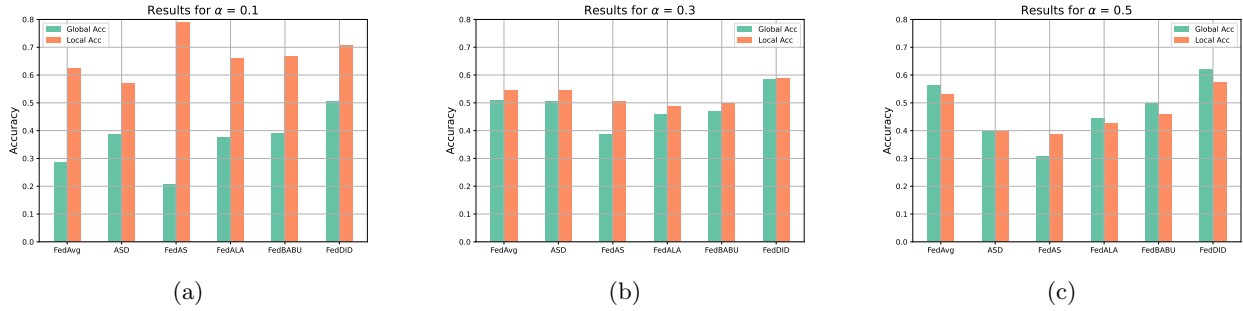

Figure 6: Average accuracy results on CIFAR-10 heterogeneity: **(a)** $\alpha = 0.1$, **(b)** $\alpha = 0.3$, and **(c)** $\alpha = 0.5$
.

## 5.5 Results on Larger Number of Clients

We study scalability under extreme partial participation on CIFAR-10 ($\alpha = 0.3$) and CIFAR-100 ($\alpha = 0.1$) by increasing to 500 clients and sampling just 1% per round. As shown in Table 3, FedDID remains the strongest overall in this setting, achieving the best global accuracy while also delivering the best local and worst-client performance. Although a few baselines obtain slightly lower global NFR, this typically comes at the cost of reduced global accuracy, indicating FedDID offers the best generalization-personalization trade-off when scaling to many clients with minimal participation.

Table 3: Results on CIFAR-10 and CIFAR-100 for 500 clients and a 1% sampling rate

| Dataset | Strategy | Avg Global Acc ↑ | Avg Global NFR ↓ | Avg Local Acc ↑ | Avg Local Class Acc ↑ | Avg Worst Client Acc ↑ | Balance ↑ |
|---|---|---|---|---|---|---|---|
| CIFAR-10 | FedAvg McMahan et al. (2017) | $0.253$ ±0.080 | $0.121$ ±0.041 | $0.348$ ±0.086 | $0.343$ ±0.082 | $0.173$ ±0.003 | 0.301 |
| | FedNTD Lee et al. (2022) | $0.317$ ±0.082 | $\mathbf{0.095}$ ±0.041 | $0.389$ ±0.089 | $0.381$ ±0.084 | $0.208$ ±0.003 | 0.353 |
| | FedAS Yang et al. (2024) | $0.218$ ±0.058 | $0.135$ ±0.049 | $0.351$ ±0.090 | $0.344$ ±0.086 | $0.171$ ±0.003 | 0.285 |
| | FedBABU Oh et al. (2021) | $0.278$ ±0.072 | $0.130$ ±0.040 | $0.374$ ±0.089 | $0.367$ ±0.086 | $0.190$ ±0.002 | 0.326 |
| | **FedDID (Ours)** | $\mathbf{0.340}$ ±0.083 | $0.101$ ±0.036 | $\mathbf{0.407}$ ±0.090 | $\mathbf{0.401}$ ±0.085 | $\mathbf{0.220}$ ±0.001 | **0.373** |
| CIFAR-100 | FedAvg McMahan et al. (2017) | $0.076$ ±0.033 | $0.038$ ±0.012 | $0.231$ ±0.064 | $0.171$ ±0.045 | $0.089$ ±0.001 | 0.153 |
| | FedNTD Lee et al. (2022) | $0.085$ ±0.033 | $0.029$ ±0.008 | $0.242$ ±0.064 | $0.185$ ±0.049 | $0.101$ ±0.003 | 0.163 |
| | FedAS Yang et al. (2024) | $0.018$ ±0.007 | $\mathbf{0.017}$ ±0.007 | $0.154$ ±0.053 | $0.097$ ±0.022 | $0.047$ ±0.000 | 0.086 |
| | FedBABU Oh et al. (2021) | $0.064$ ±0.027 | $0.034$ ±0.012 | $0.193$ ±0.055 | $0.149$ ±0.043 | $0.075$ ±0.003 | 0.129 |
| | **FedDID (Ours)** | $\mathbf{0.094}$ ±0.040 | $0.032$ ±0.010 | $\mathbf{0.250}$ ±0.067 | $\mathbf{0.203}$ ±0.056 | $\mathbf{0.111}$ ±0.002 | **0.172** |

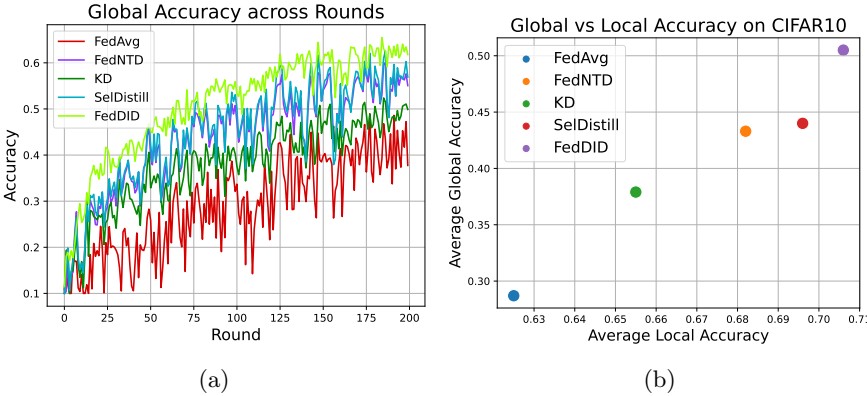

(a)  (b)

Figure 8: FedDID component comparison: **(a)** Global model performance across communication rounds and **(b)** Global-local accuracy balance.

### 5.5.1   Variations of $\beta$

We study the effect of the KL regularization weight $\beta$ by comparing several fixed settings against our adaptive formulation (indicated by Ada) on CIFAR-10. As shown in Figure 7, the adaptive $\beta$ achieves the strongest overall trade-off between personalization and generalization, delivering near-best local accuracy while also maintaining high global accuracy. Importantly, we observe a fragile zone where regularization can be harmful: moderate-to-large uniform penalties (e.g., $\beta = 2, 5$) degrade both objectives, indicating that excessive alignment can suppress useful local adaptation while still failing to improve global transfer. In contrast, our adaptive weighting strategy automatically modulates the strength of alignment to the learning regime, providing a robust balance *without* requiring manual hyperparameter tuning.

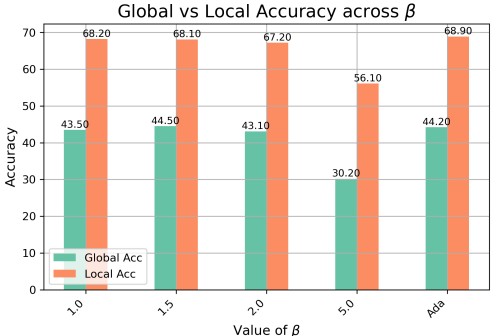

Figure 7: CIFAR-10: Effect of adaptive weight

### 5.5.2   Effect of Components in FedDID

Using CIFAR-10, we analyze each of FedDID's components. Figure 8 compares FedAvg, vanilla knowledge distillation (KD), FedNTD, FedDID's selective distillation (SelDistill), and the full FedDID model. As shown by Figure 8.a, when adding in our modified aggregation, we obtain a boost in the global test accuracy, resulting in a method that consistently surpasses other algorithms. Figure 8.b illustrates that selective distillation largely improves personalization, while the addition of our modified aggregation yields significant gains in *both* generalization and personalization.

## 6   Conclusion

In this paper, we introduce FedDID, a federated algorithm that unifies generalization and personalization. FedDID selectively aligns local and global models and adapts alignment strength using prediction-confidence drift, a signal that is justified both empirically and theoretically as a proxy for when global knowledge should be preserved versus when local adaptation should dominate. To further strengthen global transfer, we pair this with a diversity- and quality-aware aggregation that upweights clients by label diversity and model quality. Across diverse vision datasets and heterogeneity levels, FedDID consistently achieves a strong global-local trade-off, reducing global forgetting while remaining computationally lightweight. Additionally, FedDID ranks among the top methods in worst-client accuracy, highlighting improved reliability across clients. Future directions include extending this research beyond vision tasks to the language domain.

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
