# OpenReview forum: "FedDID: Discrepancy-Informed Distillation to Target Personalization and Generalization in Federated Learning"
_TMLR — Rejected by TMLR_

### Review · Reviewer_gX3Q · 2026-03-27

**Summary Of Contributions:**

This paper investigates the model regularization mechanism in federated learning. The key issue in FL algorithm is the client model drift due to data heterogeneity. The client drift without a proper regularization causes slow convergence or divergence in FL.
The paper proposed adding a regularizer that penalize the distribution drift of the local model's per-class prediction from the global model. Additioanlly, the paper uses a weighted averaging mechanism that assignes more weights on the clients with more classes rather than number of samples.
The paper provides theroetical analyses on the forgetting bound and the personalization gain. The results provide a rationale for the design of the regualrization term.
Numerical results on 5 vision datasets validates that the proposed approach enjoys higher global accuracy and comparable local accuracy.

Key strength:
1. The paper proposed a class distribution-based regularizer to regulate the client model drift. The regularization term is not gradient-base and should enjoys more theoretical benefit.

2. The paper provide sufficient description to the algorithm and explain how the experiments are conducted. The paper also reports standard deviations in the experiments.

Key weaknesses:
1. Novelty: distribution-based regularizers are not novel. E.g., FedCR [R1] and FedMDMI [R2] are using KL direvengence between the local and global model as a regularizer to mitigate data heterogeneity. The paper failed to include these papers and discuss the relationship and difference. It is hard to say whether the paper has sufficient novelty.

2. Soundness: both theoretical justifications and numerical results are not strong enough to support the algorithm design. See details below.

[R1] Zhang, H., Li, C., Dai, W., Zou, J. &amp; Xiong, H.. (2023). FedCR: Personalized Federated Learning Based on Across-Client Common Representation with Conditional Mutual Information Regularization. <i>Proceedings of the 40th International Conference on Machine Learning</i>, in <i>Proceedings of Machine Learning Research</i> 202:41314-41330 Available from https://proceedings.mlr.press/v202/zhang23w.html.

[R2] Hao Zhang, Chenglin Li, Nuowen Kan, Ziyang Zheng, Wenrui Dai, Junni Zou, and Hongkai Xiong. 2024. Improving generalization in federated learning with model-data mutual information regularization: a posterior inference approach. In Proceedings of the 38th International Conference on Neural Information Processing Systems (NIPS '24), Vol. 37. Curran Associates Inc., Red Hook, NY, USA, Article 4341, 136646–136678.

**Audience:**

No

**Audience Explanation:**

1. The paper lacks sufficient novelty. The KL divergence has already been used as a regularizer in FL algorithm design.
2. The paper lacks sufficient theoretical results or novel proving techniques. Most of the derivations are straightforward. The proof of Lemma 2 has approximations (e.g., eq(53)).
3. The numerical results does not match SOTA benchmark results, so it is hard to determine whether the algorithm is useful or not.

**Claims And Evidence:**

No

**Claims Explanation:**

1. The proposed method lacks solid theoretical justification. Lemma 2 is a trivial derivation from Assumptions 1 and 2. No further convergence analysis or performance guarantee is provided for the algorithm.
2. The numerical results are not matching existing benchmark results. From existing benchmarks, e.g., [R3-R5], the accuracy of the algorithms are under-reported. E.g., from [R5], when $\\alpha=0.1$, the averages accuracy of un-participated clients (i.e., global model) still maintains ~0.5 accuracy for FedVag, but the paper reports accuracy < 0.3.
3. The std in the numerical results a pretty large, showling insufficient statisitical significance. E.g., in table 2 for CIFAR-100, FedNTD has accuracy $0.272\\pm 0.081$ and FedDID is $0.278\\pm 0.080$. Claiming the proposed method outperforms existing method is not convincing.

[R3] Wenke Huang, Mang Ye, Zekun Shi, Guancheng Wan, He Li, Bo Du, and Qiang Yang. 2024. Federated Learning for Generalization, Robustness, Fairness: A Survey and Benchmark. IEEE Trans. Pattern Anal. Mach. Intell. 46, 12 (Dec. 2024), 9387–9406. https://doi.org/10.1109/TPAMI.2024.3418862

[R4] Zhang, H., Li, C., Dai, W., Zou, J. &amp; Xiong, H.. (2023). FedCR: Personalized Federated Learning Based on Across-Client Common Representation with Conditional Mutual Information Regularization. <i>Proceedings of the 40th International Conference on Machine Learning</i>, in <i>Proceedings of Machine Learning Research</i> 202:41314-41330 Available from https://proceedings.mlr.press/v202/zhang23w.html.

[R5] Daoyuan Chen, Dawei Gao, Weirui Kuang, Yaliang Li, and Bolin Ding. 2022. PFL-bench: a comprehensive benchmark for personalized federated learning. In Proceedings of the 36th International Conference on Neural Information Processing Systems (NIPS '22). Curran Associates Inc., Red Hook, NY, USA, Article 679, 9344–9360.

**Requested Changes:**

1. The paper defines $J$=# classes, but the summations are from $j=0$ to $J$ of total $J+1$ classes.

2. Discuss the difference between the proposed method and existing distribution-based FL algorithms

---

> ### Author Response · Authors · 2026-05-28
> **Response to Reviewer gX3Q**
>
> We thank the reviewer for their feedback and address their concerns below.
>
> 1. $\textbf{Novelty and Additional Comparisons:}$ In the original submission, we listed how our method was different from competing methods that make use of knowledge distillation in Section 2.2. In summary, FedDID differs from prior KD-based FL approaches by making distillation classwise and adaptive rather than uniform or globally weighted. Existing methods use KD to preserve global knowledge, align mutual predictions, and/or guide personalization, but they generally apply the distillation signal uniformly across classes or clients. In contrast, FedDID adjusts the strength of distillation based on local-global prediction confidence discrepancies, allowing each client to selectively preserve the global knowledge that is most relevant to its local distribution. This enables FedDID to better balance global generalization with local personalization under heterogeneous data.
> We also appreciate the reviewer for bringing to our attention additional prior literature. In the revision, we have referenced both FedCR and FedMDMI in Appendix D and provide a detailed summary on how FedDID differs from these two approaches. Specifically, FedDID differs from FedMDMI and FedCR in that it does not simply encourage the preservation of shared global or cross-client information. These methods do not explicitly decide whether a given global prediction is relevant for a specific client or class. FedDID instead uses local-global confidence discrepancies to selectively adjust classwise distillation, preserving useful global knowledge while avoiding unnecessary constraints on client personalization. To demonstrate this, we have compared against FedMDMI in this section (as it is the most recent of the two approaches) across CIFAR-10/100 and TinyImageNet. As indicated by Table 4 and Figure 9 in the appendix, FedDID excels over FedMDMI at both the global and local level on all datasets.
>
> 2. $\textbf{Numerical Results:}$ It is common in other papers when reporting accuracy to take the final communication round results, average this across seeds, and report this. For the referenced papers, our understanding is that the reported “global accuracy” is based on a single model checkpoint near the end of training. In contrast, our tables report the average accuracy attained across communication rounds, averaged over three seeds. This choice is intentional: our work is interested in overall the stability of global and personalized performance throughout federated training under heterogeneity. In practical FL deployments, models may be selected, evaluated, or deployed at different communication budgets, and unstable trajectories can lead to poor intermediate performance even if the final checkpoint is strong. To avoid ambiguity, we rephrased the evaluation section to better reflect this. Additionally, to allow better understanding of how this relates to final accuracy, we have placed the averaged global accuracy curves across all communication rounds in Appendix F.1. Comparing these plots to, for instance, CIFAR-100 with the FedCR paper, we see similar final accuracy results. Furthermore, the larger standard deviation (across all algorithms) reflects variation in the entire training trajectory, including differences in early-round convergence speed and fluctuations, rather than only variation at the final model. This makes the metric stricter but can naturally produce larger deviations. This can again be verified when looking at the global accuracy curves in Appendix F.1 (where the shaded regions around the curves indicate standard deviation), showing our method overall has relatively small variance at individual communication rounds and maintains a consistently strong trajectory.
>
> 3. $\textbf{Theoretical Results:}$ We agree that Lemma 2 is not intended to serve as a full theoretical convergence or performance guarantee. Rather, its purpose is to provide a simple analytical explanation of when the adaptive KD term can improve the local personalization objective via the regularization gradient being sufficiently aligned with the local cross-entropy gradient. We have revised the text to clarify that this lemma is a result motivating the mechanism, not the main theoretical guarantee. In addition, the manuscript already includes a convergence analysis in Appendix B, which studies the optimization behavior of the proposed objective under standard assumptions, adding to the theoretical rigor of the paper. This is already referenced at the beginning of Section 4.3.
>
> 4. $\textbf{Equation change:}$ We thank the reviewer for pointing out this inconsistency. As such, we have now changed the summation equation to reflect this correction.

---

### Review · Reviewer_hWDD · 2026-05-04

**Summary Of Contributions:**

This paper proposes FedDID (Federated Discrepancy-Informed Distillation), a federated learning method that simultaneously improves global generalization and local personalization through adaptive class-level knowledge distillation. The core idea is to use the per-class confidence drift between local and global models as a signal to adaptively regulate the strength of knowledge distillation—applying stronger regularization to classes with large drift while allowing more local adaptation for classes with small drift. Additionally, the paper introduces a weighted aggregation strategy that combines client label diversity and model quality. Theoretical analysis (convergence, forgetting upper bound, personalization gain) and experimental validation on five image classification datasets are provided. The main strengths include a clear motivation using confidence drift as a unified signal for both generalization and personalization, a simple and elegant design, comprehensive comparisons against 13 baselines, a “Balance” metric that facilitates fair evaluation of the global–local trade-off, negligible communication overhead for practical use, and adequate ablation studies and β-sensitivity analysis. The main weaknesses are that the experimental scale is too small (limited datasets, model architectures, and number of clients).

**Audience:**

Yes

**Audience Explanation:**

Simultaneously addressing generalization and personalization in federated learning is an important and active research direction. The idea proposed in this paper—using confidence drift to guide class-level distillation strength—is indeed novel and practical, and offers valuable insights for researchers working on federated learning, knowledge distillation, and non-IID data processing. However, due to the limited scale of experimental validation, the contributions of this work remain closer to the proof-of-concept stage, and its practical guidance for real-world deployment scenarios is limited.

**Broader Impact Concerns:**

The paper does not include a Broader Impact Statement.

**Claims And Evidence:**

No

**Claims Explanation:**

1. The dataset scale is too small to support the claim of "strong empirical performance". CIFAR-10/100, TinyImageNet (64×64), BloodMNIST, and OrganCMNIST are all extremely small-scale datasets. In current federated learning research, these datasets are no longer sufficient to fully validate a method's effectiveness. The lack of validation on larger-scale, more realistic datasets (e.g., large-scale FEMNIST, Landmarks, iNaturalist, or federated datasets in text/speech domains) casts doubt on the generalizability of the conclusions.

2. The model architecture is overly simple. All experiments use only a simple CNN with 2 convolutional layers and 2 fully connected layers. Modern federated learning research typically validates on ResNet-18 or similar architectures at a minimum. Performance gains obtained on such a simple model do not guarantee the same on more complex models — especially because the effectiveness of knowledge distillation is highly dependent on model capacity.

3. Results on TinyImageNet and CIFAR-100 show that the method's advantage diminishes when the number of classes is large. The paper acknowledges this but only calls it a "modest degradation," yet this is precisely the real-world scenario where improvement is most needed.

**Requested Changes:**

1. Expand the experimental scale. All current datasets (CIFAR-10/100, TinyImageNet 64×64, MedMNIST series) are insufficient in scale and complexity to support publication-level conclusions. It is recommended to at least add: (a) larger-scale/higher-resolution datasets (e.g., ImageNet subsets, Landmarks-User-160k, iNaturalist, etc.); (b) validation on non-image domains (e.g., federated scenarios for NLP tasks) to assess generalizability; (c) datasets that better reflect real-world federated settings (e.g., large-scale FEMNIST from the LEAF benchmark, Reddit, and other naturally heterogeneous datasets).

2. Use more complex model architectures. At a minimum, validation should be performed on ResNet-18 or MobileNet-level architectures. The current 2-layer CNN is overly simple and cannot reflect real-world deployment requirements, nor can it confirm whether the method remains effective on deeper networks. The behavior of knowledge distillation may differ fundamentally between shallow and deep networks.

---

> ### Author Response · Authors · 2026-05-28
> **Response to Reviewer hWDD**
>
> We thank the reviewer for their feedback. Below, we address their comments and concerns.
>
> 1. $\textbf{Dataset Scale:}$ We agree with the reviewer that the experimental scale can be expanded. Therefore, we evaluate on 2 additional large, naturally heterogeneous datasets: FEMNIST and Fed-ISIC (to align with our evaluations on the medical MedMNIST datasets). These results are presented in Appendix F.3. As indicated by these results in Table 6 in Appendix F.3, On FEMNIST, FedDID achieves the best global accuracy, best local accuracy, best local class accuracy, and best balance score, while maintaining competitive NFR. On Fed-ISIC, where all methods perform more similarly due to the smaller number of clients and moderate sampling rate, FedDID still attains the best global accuracy and best overall balance. These results show that FedDID’s gains are not limited to synthetic Dirichlet partitions, but also extend to naturally heterogeneous federated datasets with client distributions induced by users or acquisition sites. In total, we now present extensive evaluation on $\textbf{seven}$ datasets spanning standard image benchmarks, medical image benchmarks, and naturally heterogeneous federated datasets. We have not added NLP experiments because the scope of this work, as stated in the introduction, is image classification under heterogeneous FL, and our method is designed and evaluated in that setting. The architectures, benchmark protocols, and heterogeneity regimes considered in this work are specific to visual classification tasks. While the broader principle of discrepancy-aware distillation may extend to NLP, doing so would require selecting appropriate NLP benchmarks for federated learning and adapting the distillation formulation to the target NLP setting. For example, text classification may still use class-level distillation, whereas language modeling or sequence generation may require token-level distillation. We therefore view extension to NLP as an important but separate future direction, and we now mention this in the conclusion of the revised manuscript.
>
> 2. $\textbf{Model architecture:}$ We have now added results using a pretrained ResNet-18 in Appendix F.4 to evaluate whether FedDID remains effective with a deeper architecture. As shown in Table 7 in Appendix F.4, FedDID continues to obtain the best overall balance of generalization and personalization. In particular, FedDID achieves the highest global accuracy among all methods while also matching the strongest personalized baselines at the local level. Although FedAS and FedALA attain slightly higher local accuracy or worst-client accuracy, they do so with substantially weaker global performance. In contrast, FedDID preserves strong local performance while improving global generalization, leading to the best balance score. These results suggest that FedDID’s discrepancy-aware distillation remains effective beyond shallow CNNs, echoing our conclusions from Table 1.
>
> 3. $\textbf{TinyImageNet/CIFAR-100 Results:}$ We agree that large class spaces are an important and challenging setting, and we have revised the discussion to more clearly reflect this difficulty. However, the results do not indicate that FedDID’s advantage disappears as the number of classes increases. On CIFAR-100, FedDID achieves the best global accuracy, matches the best balance score, and remains highly competitive in local accuracy, demonstrating that the method continues to provide strong generalization and personalization in a larger class space. On TinyImageNet, FedDID achieves the second-best balance score and remains competitive with the strongest baseline, FedNTD, while still outperforming many other FL and pFL methods. At the same time, we acknowledge that TinyImageNet is a more difficult regime, where several methods either degrade substantially or fail to train reliably. This suggests that the reduced margin is not specific to FedDID, but reflects the broader difficulty of maintaining both global generalization and local personalization when the class space becomes large and client data are highly heterogeneous. We have therefore revised the text to avoid understating this challenge and to frame large-class federated learning as an important direction for future work, rather than simply as a modest degradation.

---

### Review · Reviewer_56LD · 2026-05-17

**Summary Of Contributions:**

This paper focuses on federated learning under statistical heterogeneity, where non-iid data across clients degrades both global generalization and local personalization. It introduces FedDID (Federated Discrepancy-Informed Distillation), which pursues both objectives by adaptively weighting global-to-local knowledge distillation on a per-class basis. The weighting is performed based on the discrepancy in class prediction confidence between the local and global model, which is considered as an informative statistic to identify classes that are likely to be forgotten, as well as to promote personalization. Essentially, a local model is penalized harder for drifting away from the global model on classes where their predictions disagree the most. The approach is also complemented by a modified aggregation strategy that favors clients with higher accuracy and greater label diversity. Overall, the method yields strong global and local performance without much computational overhead.

**Additional Comments:**

- The proposed model aggregation strategy assumes safe-sharing of the local scalar per client \gamma_k, and the paper claims that this still maintains client privacy as it does not expose client per-class counts. Although this sounds reasonable, in principle this scalar encodes the proportion of classes present in the client's local data. In an iterative adversarial probing scenario, could repeated observations of this scalar potentially leak information about the client’s data distribution? What do the authors think about this?

**Audience:**

Yes

**Audience Explanation:**

I believe the research community interested in federated learning would be interested in the proposed approach.

**Claims And Evidence:**

No

**Claims Explanation:**

- The paper claims minimal computational overhead with respect to existing methods, although Section 5.3.2 and Figure 4 are rather brief on elaborating this. Firstly, this analysis is only performed on CIFAR-10, rather than the large-scale simulations that are already present in the paper (e.g., how is it for the TinyImageNet simulations from Table 2?). Secondly, even for CIFAR-10, the strongest competitor approaches from Table 1 are not considered for the “preferable computational overhead” claim (e.g., It would be good to see strong local-model performant methods FedAS, Ditto, FedPer, also in Figure 4).

- The backbone architectures considered for each dataset and method is not clear in the manuscript. The appendix defines some choices, but is the same architecture considered for all compared approaches? How reliable are the results with respect to the capacity of the backbone model scale?

- Some results in Table 2 TinyImageNet need to be revisited. There are two rows where global NFR is 0.000 (i.e., no negative flips) and Acc is 0.005 (i.e., chance-level for TinyImageNet). This seems more like a dead training run artifact rather than reliable results, which should be clarified.

**Requested Changes:**

- The empirical performance is rather weak and unstable across several benchmarks. There seems to be a lot of averaging in the main results presented in Tables 1 & 2. When the accuracies are averaged across classes and seeds, and also the global and local accuracies are averaged, the Balance column looks fine. However, worst case performance would possibly drive the actual utility of the algorithm. Therefore, I would like to ask if these tables could also be presented in terms of the worst case performance observed across 3 repetition seeds, rather than averages?

- Regarding my comment above on the claims regarding the computational overhead, I would like to ask additional overhead comparisons for the TinyImageNet simulations from Table 2, and also the strong local-model performant methods FedAS, Ditto, FedPer, in Figure 4.

- How reliable are the results with respect to the capacity of the backbone model architecture scale (e.g., if a larger architecture was considered)?

- Please clarify the ambiguous TinyImageNet results from Table 2, as mentioned in my comment above.

---

> ### Author Response · Authors · 2026-05-28
> **Response to Reviewer 56LD**
>
> We thank the reviewer for their feedback and address their concerns below.
>
> 1. $\textbf{Computational overhead:} $ We agree that adding additional pFL methods to the communication cost comparison for CIFAR-10 is beneficial. Therefore, we have modified Figure 4 to reflect this concern, adding FedALA, FedAS, and Ditto for comparison. As indicated by this modified figure, the personalized methods tend to have the best wall-clock time for CIFAR-10 (even beating the standard FedAvg algorithm). We note, however, that our method presents modest cost compared to these pFL algorithms, particularly coupled with the fact that our method performs well on both global and local objectives (whereas pFL methods specialize in personalization). Additionally, we have provided a computational cost comparison on FEMNIST (which is a much larger dataset than TinyImageNet) in Appendix Section F.5, referenced by Figure 13. As indicated by this newly added section, FedDID still is comparable in communication cost to FedAvg, with the pFL methods now ranking much higher in communication cost upon the addition of hundreds of more clients with this dataset. Based on FedDID’s substantial improvements over other methods at both the global and local level on this dataset (Table 6), FedDID achieves these gains while maintaining communication cost comparable to FedAvg and FedNTD and far below FedALA and Ditto.
>
> 2.  $\textbf{Architecture choices:}$ The same architecture is utilized across all methods; the results in the main text refer to the simple CNN described in the Appendix. We have also added additional experiments for a larger model (Resnet-18) in Appendix F.4. We find that even across this larger architecture, FedDID obtains the best balance of generalization and personalization.
>
> 3. $\textbf{TinyImageNet:}$ We thank the reviewer for pointing this out. TinyImageNet rows with global accuracy near chance level and NFR equal to 0 should not be interpreted as meaningful evidence of stable retention. Since NFR measures negative flips relative to previously correct predictions, a model that fails to learn the task can trivially produce a very low or zero NFR because there are few correct predictions to lose. Thus, these entries reflect degenerate or failed training behavior for those baselines under the TinyImageNet setting. Accuracy and NFR are best viewed together to understand whether a 0 NFR reflects retention or is just an artifact of failed training. In the revision, we explicitly mark these cases as failed runs and clarify that NFR is only meaningful when paired with nontrivial accuracy. This clarification does not affect the main conclusion, since on TinyImageNet, FedDID maintains nontrivial global accuracy while also achieving competitive NFR.
>
> 4.  $\textbf{Worst-case performance:}$ We agree that worst case performance is highly informative. The initial submission presented a visual comparison of average vs worst local accuracy in Section 5.3.3 on three datasets across three different seeds. The results in this section show that FedDID maintains strong average accuracy while achieving the second-best worst-client accuracy (notable because FedDID is not a dedicated pFL baseline like FedALA). We extend these results to Appendix F.2, where we numerically report the worst global and local performance for all datasets. In Appendix F.2, we observe that FedDID remains highly competitive with pFL baselines in terms of worst-local accuracy and excels in providing a high worst-global accuracy over all other baselines.
>
>  $\textbf{Aggregation strategy question:}$ Our scalar $\gamma_k$ encodes a mixture of the client’s training accuracy and label diversity (computed locally, prior to sharing with the server). Thus, while the scalar may reveal a coarse signal about whether a client’s data is broadly diverse or highly skewed, it does not provide enough information to reconstruct the client’s per-class distribution in general. That said, we agree that under a stronger iterative probing threat model, repeated observations of $\gamma_k$ could potentially leak limited coarse information about changes in a client’s label support. Therefore, in privacy-sensitive deployments, mechanisms like differential privacy noise could be used to further secure this scalar.

---

### Author Response · Authors · 2026-05-28
**Summary of Changes in Revised Manuscript**

We thank all reviewers for their time and feedback. We have uploaded our revised manuscript based on reviewer feedback. Below, we have written a summary of the main changes now present within the revised manuscript:

1. Expanded the communication cost comparison in Figure 4 to include additional personalized FL baselines, including FedALA, FedAS, and Ditto.

2. Added a new FEMNIST communication cost analysis in Appendix F.5 / Figure 13 to evaluate scalability on a larger federated dataset.

3. Added experiments on two naturally heterogeneous federated datasets, FEMNIST and Fed-ISIC, reported in Appendix F.3 / Table 6.

4. Added experiments with a pretrained ResNet-18 architecture in Appendix F.4 / Table 7 to evaluate whether FedDID remains effective beyond the simple shallow CNN setting.

5. Marked TinyImageNet cases with near-chance global accuracy and zero NFR as failed runs, and clarified that NFR should only be interpreted alongside nontrivial accuracy.

6. Added numerical worst-case global and local performance results for all datasets in Appendix F.2.

7. Revised the CIFAR-100 and TinyImageNet discussion to better acknowledge the difficulty of larger class spaces under heterogeneous FL, and framed large-class federated learning as an important direction for future work.

8. Added FedCR and FedMDMI to the related discussion in Appendix D, and clarified how FedDID differs from these methods through classwise, discrepancy-informed adaptive distillation.

9. Added an empirical comparison against FedMDMI in Appendix D, with results reported in Table 4 and Figure 9.

10. Rephrased the evaluation section to clarify that reported global accuracy is averaged across communication rounds and then averaged over three seeds, rather than taken only from the final communication round.

11. Added averaged global accuracy curves across communication rounds in Appendix F.1 to contextualize final accuracy trends and trajectory-level stability.

12. Clarified that the reported standard deviations reflect variation across the full training trajectory, including early-round convergence and fluctuations, rather than only final-checkpoint variation.

13. Revised the theoretical discussion to clarify that Lemma 2 motivates the adaptive KD mechanism rather than serving as a full theoretical guarantee, while noting that the convergence analysis is provided separately in Appendix B.

14. Corrected the reviewer-identified summation equation/notation inconsistency.

---

### Decision · Action_Editor_46JC · 2026-06-23

**Recommendation:** Reject

**Additional Comments:**

For a possible resubmission, the authors should take into account additonal comments of reviewer 56LD:

- Unfortunately I did not understand why new FEMNIST experiments appeared in response to the question about TinyImageNet computational cost experiments (which were already performed for the original submission), with the statement that "it is a much larger dataset than TinyImageNet". Firstly, I was asking more on the difficulty of the classification task itself, when the dimensionality of the data increases. Secondly, it is not clear how FEMNIST is a more challenging task than TinyImageNet, and the authors do not elaborate their perspective quite well.

- As a suggestion for later revisions, perhaps the authors should include a transparent discussion on the limitation concerning the sharing of the per-client local scalar.

**Audience:**

Yes

**Audience Explanation:**

The method is of potential interest for researchers in the area of federated learning.

**Claims And Evidence:**

No

**Claims Explanation:**

The main results in Table 2 contain failed training runs of baseline methods. In the supplement, it is stated that baseline methods were trained with hyperparameters stated in the original papers. This indicates inappropriate benchmarking. A comparison should include a fair and transparent hyperparmeter tuning of baselines. If unavoidable, failed runs should be analyzed and discussed (e.g. if a method is not suited for the task).
The statement that FedDID adds negligible communication overhead is not supported by the data, see revised Figure 4.
Novelty with respect to previous methods is not sufficiently discussed in the main text.

**Resubmission Of Major Revision:**

The authors may consider submitting a major revision at a later time.